# Local Distributed Node for Power Quality Event Detection Based on Multi-Sine Fitting Algorithm

**DOI:** 10.3390/s24082474

**Published:** 2024-04-12

**Authors:** Domenico Luca Carní, Francesco Lamonaca

**Affiliations:** Department of Computer Engineering, Modeling, Electronics and Systems, University of Calabria, 87046 Rende, Italy; dl.carni@dimes.unical.it

**Keywords:** measurement, monitoring system, distributed monitoring system, power signal event, power quality, sinusoidal signal alteration detection

## Abstract

The new power generation systems, the increasing number of equipment connected to the power grid, and the introduction of technologies such as the smart grid, underline the importance and complexity of the Power Quality (PQ) evaluation. In this scenario, an Automatic PQ Events Classifier (APQEC) that detects, segments, and classifies the anomaly in the power signal is needed for the timely intervention and maintenance of the grid. Due to the extension and complexity of the network, the number of points to be monitored is large, making the cost of the infrastructure unreasonable. To reduce the cost, a new architecture for an APQEC is proposed. This architecture is composed of several Locally Distributed Nodes (LDNs) and a Central Classification Unit (CCU). The LDNs are in charge of the acquisition, the detection of PQ events, and the segmentation of the power signal. Instead, the CCU receives the information from the nodes to classify the PQ events. A low-computational capability characterizes low-cost LDNs. For this reason, a suitable PQ event detection and segmentation method with low resource requirements is proposed. It is based on the use of a sliding observation window that establishes a reasonable time interval, which is also useful for signal classification and the multi-sine fitting algorithm to decompose the input signal in harmonic components. These components can be compared with established threshold values to detect if a PQ event occurs. Only in this case, the signal is sent to the CCU for the classification; otherwise, it is discarded. Numerical tests are performed to set the sliding window size and observe the behavior of the proposed method with the main PQ events presented in the literature, even when the SNR varies. Experimental results confirm the effectiveness of the proposal, highlighting the correspondence with numerical results and the reduced execution time when compared to FFT-based methods.

## 1. Introduction

The new development of power grids with the integration of new-generation systems such as the green micro-power generation systems, together with the increase in the number of nonlinear loads and power electronics equipment, increases the importance of the Power Quality (PQ) evaluation [1,2,3]. Indeed, the quality of the power supply can change over time due to several reasons such as atmospheric phenomena and the switching of heterogeneous loads that can introduce alterations in the nominal shape of the current, voltage, and frequency. All these can generate a PQ event. The PQ event degrades the device’s functionality supplied by the power signal [4], generating unwanted operative conditions, device malfunction, service interruptions, and dangers to human health and devices. To ensure an established level of PQ, it is necessary to identify the PQ event source to take proper decisions and reactions [5]. To achieve this goal, it is necessary to classify PQ events. A specific class of events can be generated by specific sources. The event class depends on its time-variant statistical characteristics [6], and the categories of the PQ events are described in IEEE Standard 1159 [7]. Instead, the Standard IEC61000-4-30 [8] reports the PQ measurement methods to achieve the information to obtain the class of an event and its parameters.

The PQ evaluation is typically performed after a problem occurs by a trained operator. The reliability of the evaluation strongly depends on the operator’s ability and experience. Moreover, this approach does not permit preventive actions. Furthermore, the probability that a PQ event occurs during the operator’s observation time can be low, making the monitoring ineffective. For these reasons, recently, several devices for the automatic detection and classification of PQ events have presented in the literature [9,10,11]. The general architecture [12] is shown in Figure 1, and it is composed of the blocks: signal acquisition and segmentation, feature extraction, classification, and decision-making.

The relevant cost of the devices based on this architecture mainly depends on the computational requirements of the classifications block. High costs limit the scalability of the monitoring system and its capillary distribution over the whole power network. Moreover, due to the source of the PQ events, they can occur randomly, and it is not always possible to define an exact time in which an event will occur [13]. For this reason, the monitoring and classification system will, most of the time, do nothing except wait for an event, as long as an event happens, and its significant computing power will remain practically unused for almost all of the time.

It is worth noting that not all the operations need to be executed at the measurement point. In particular, only the detection and acquisition of the PQ event must be performed at the measurement point; the classification can be performed by a remote device or in the cloud.

For this reason, in this paper, a new Automatic PQ Events Classifier (APQEC) architecture is proposed. It is made up of two main component sets: the first is a set of Locally Distributed Nodes (LDNs) that detect, segment, and acquire the portion of the Power Signal (PS) affected by the PQ event. These data are then transmitted to the second set of Central Classification Units (CCUs) which perform the PQ event classification. The advantage of this architecture is that the LDN can be implemented with low-cost devices, and they are the only ones that need to be distributed, while each CCU can manage multiple LDNs, thus reducing the overall cost of the proposed monitoring system. The improvement of the proposed solution is highlighted when the number of measurement points increases. Considering that tools such as Power Quality Analyzers or PCs and hardware for the implementation of the methods proposed in the literature cost more than a thousand Euros, this cost must be multiplied by the measurement points. Instead, in the proposed solution, a single PC allows you to monitor multiple measurement points with LDNs, which cost a few tens of Euros.

The scientific novelty of this approach is that a remote low-cost and low-capability subsystem triggers the local high-capability subsystem, which can be used in completely different areas than PQ, such as intrusion detection and classification, telecommunication signal detection and classification, Structural Health Monitoring systems, and so on.

From the analysis of the recent literature [14,15,16,17,18], the proposed methods perform the detection as the result of the classification or use of mathematical tools that are computationally expensive (e.g., the FFT has a computational complexity **O**(n log(n))) and, therefore, all of them are not suitable for implementation on HW with reduced computational capabilities. As a consequence, since the aim of the paper is to reduce the monitoring costs by proposing an LDN with limited computational resources, a new detection and segmentation method characterized by a lower computational burden is proposed. The proposed detection method uses the multi-sine fitting algorithm [19] to recognize the presence of PQ events. Unlike [19], a Sliding Analysis Window (SAW) is used to permit the analysis of a time-varying signal and determine the input of the multi-sine fitting algorithm. In particular, the events are detected as an increase in specific harmonic components in the SAW. This method permits online detection also by using low-cost boards such as Arduino. This is possible owing to the low computational complexity of the multi-sine fitting algorithm used to evaluate the harmonic components if it is compared with the Fourier Transform. Once a PQ event is acquired, the node sends the acquired signal to the CCU where it is elaborated for the event classification and the evaluation of the characteristic parameters. Then, any corrective action can be evaluated and executed. The CCU can be implemented in a server or in the cloud. The CCU typically has a higher computational capability and can execute the promising classification procedure proposed by the authors in [20] or others available in the literature. It is good to note that the proposed solution does not have the purpose of minimizing the data transferred to the processing center by increasing the capability and the computation performed in the node. As a consequence, according to [21], it cannot be considered edge computing. Conversely, it is proposed to divide a measurement instrument into two different systems. The first one can be replicated and distributed over many measurement points, and it does not perform complex processing on the acquired data, but the second one processes the data to perform the PQ event analysis.

This paper is organized as follows. In Section 2, the proposed architecture of an APQEC is described. In Section 3, the functions of the CCU are introduced, and the possible implementations are summarized. In Section 4, the proposed LDN is described, and the algorithm for the detection, segmentation, and acquisition is analyzed. In Section 5, the results of the numerical tests are reported to determine the effectiveness and the performances achievable by the proposed method. In Section 6, experimental results are presented to confirm the numerical ones and evaluate the performances achievable on a low-computational capability and low-cost node such as Arduino. The conclusions follow.

## 2. Proposed Distributed APQEC

The general block diagram of the architecture of an APQEC is represented in Figure 1 [12]. The Signal acquisition and segmentation block acquires the power signal. It operates to detect if a PQ event occurs and, in this case, stores the non-stationary part of the input signal related to the event. This detection can be obtained by using wavelet or Fourier transforms. In the feature extraction block, from the acquired samples are evaluated features highlighting the characteristics of the PQ event, allowing for the classification. The characteristics that must be extracted depend on the classifier implemented in the classification block, and, therefore, they can be obtained by different mathematical tools. Finally, the classification and decision-making blocks are typically implemented by using artificial intelligence techniques such as Artificial Neural Networks [22,23], Support Vector Machines [24,25,26,27], and deep learning techniques [20,28,29,30,31,32,33].

For each point of the power net to be monitored, this architecture requires the use of devices with considerable computing resources, essentially needed for feature extraction and classification. These actions must be exploited only in the case where a PQ event is detected. It means that the full computational capabilities can be used only a few times or never. Based on this consideration, to reduce the cost of the APQEC, it is proposed to divide the monitoring node in two devices. The first device is the LDN shown in Figure 2, and it is devoted to acquiring the signal, detecting an event, and, in the case an event is detected, performing its segmentation, and transmitting the acquired sample to the second device. The second device of the proposed system is the CCU shown in Figure 3, which receives the data from one or more LDNs, extracts the PQ event features, and performs its classification, for proposing or implementing any corrective actions.

## 3. Central Classification Unit

To design the CCU, the recent literature was analyzed, and different solutions for the automatic classification of the PQ events [14,34,35] were studied. In this field, the use of the Short-Time Fourier Transform (STFT) is proposed in [36,37,38,39]. The non-stationary nature of the PQ event increases the uncertainty in the classification due to the limited time and frequency resolution typical of the STFT. The use of multi-resolution fuzzy and S-Transform classification systems is proposed in [40]. This solution requires the definition of a variable time window that must be established, but the classification accuracy can be improved. The use of an S-transform and a Convolution Neural Network (CNN) is proposed in [41,42], but the results show a miss-classification problem. A Support Vector Machine classifier is proposed in [43] with a good classification accuracy, but also in this case, some miss-classifications must be resolved. The authors in [20] proposed a classifier characterized by a higher classification accuracy. This classifier is based on the use of the Huang–Hilbert Transform (HHT) and a CNN. The higher classification accuracy arises from the robustness of the HHT in the analysis of the non-stationarity signal and of the CNN to operate on noisy data. For the evaluation of the parameters of the event, in [44], the authors propose a tool based on the analysis of Intrinsic Mode Functions (IMFs) resulting from the EMD [45,46,47]. This decomposition is the basis through which the HHT is evaluated, so the same advantages of the HHT are provided by this procedure. For these reasons in the paper, this method is considered for the implementation of the CCU.

An important aspect that is highlighted by this analysis is that, regardless of the classification method used, the latter must receive a constant number of samples that contain the PQ event together with part of the signal before the event occurs. This aspect serves to establish how the output of the segmentation block should be formed.

## 4. Locally Distributed Node Algorithm

To develop detection and segmentation algorithms suitable for low-cost hardware of the LDN, the typical methodologies proposed in the literature cannot be used. The solutions based on wavelet transform [48,49], STFT, and the auto-regressive model [50] require high computational power, and, as a consequence, expensive hardware. In the design of the detection method algorithm, the PQ events taken into consideration are the most typical ones defined by international standards and in the literature [14,22,23,24,25,26,34,35,36,37,38,39,51]. In particular, the PQ events considered are as follows: interruption, sage, swell, harmonic, transient, flicker, notch, and spike. The proposed detection and segmentation of PQ events method is schematized in Figure 4. Each acquired sample is stored in a circular buffer implementing a FIFO policy. The logic of the circular buffer is to keep the new samples from overwriting the oldest ones. The Buffer Manager (BM) takes the SAW, a subset of the circular buffer, to be analyzed to detect an event. The samples in the SAW are filtered with a low-pass digital filter with a cut-off frequency of 200 Hz. The three-parameter multi-sine fitting algorithm is executed on the filtered samples. Filtering is necessary to reduce the influence of noise and make the proposed method suitable for noisy environments. The filtered samples are processed by a multi-sine fitting algorithm. This is a non-iterative procedure that estimates the parameters of the fundamental and harmonic components by minimizing the RMS error between the samples received in the input and the reconstructed ones [52]. In this way, the distortion of the nominal sinusoidal caused by a PQ event can be detected. If a PQ event is detected, the BM performs the segmentation with the samples stored in the circular buffer to the CCU for the classification of the event. Otherwise, the BM selects the next subset of the memory to be analyzed, starting from a pre-established shift in the circular buffer memory location. The dimension of the SAW is chosen according to the number of samples necessary for subsequent classification.

### 4.1. Three-Parameter Multi-Sine Fitting Algorithm

Theoretically, the power signal can be modeled as a sinusoid with an amplitude and frequency depending on the country. The PQ events introduce unwanted frequency components; then, by estimating these components, it is possible to detect this last. With this aim, the input signal is modeled as follows:(1)$$s\left(t\right)=\sum_{p=1}^{P}A_{p}\sin\left(2\pi pft\right)+\sum_{p=1}^{P}B_{p}\cos\left(2\pi pft\right)+D$$
where $A_{p}$ and $B_{p}$ are the in-phase and in-quadrature components of the *p*-th harmonic, *f* is the frequency of the signal, *P* is the number of harmonic components of the signal, and *D* is the offset. The estimation of the $A_{p}$, $B_{p}$, and *D* values can be obtained by using the multi-sine fitting algorithm proposed in [19,52]. The algorithm considers the vector representation of the *M* samples of the input signal s(t) acquired with a sampling period $T_{c}$, and the variable vector *x*, which are the harmonics parameters of (Equation 1):(2)$$x={\left[A_{1},B_{1},A_{2},B_{2},\cdots ,A_{P},B_{P},D\right]}^{T}$$

By this, (Equation 1) can be rewritten, by considering the vector *s* of s(t), as follows:(3)s=Gx
where the matrix G is defined as follows:(4)$$\begin{array}{c} G=\\ \;\;\;\;\;\;\left[\begin{array}{cccccccc} S_{1,1} & C_{1,1} & S_{2,1} & C_{2,1} & \cdots & S_{P,1} & C_{P,1} & D\\ S_{1,2} & C_{1,2} & S_{2,2} & C_{2,2} & \cdots & S_{P,2} & C_{P,2} & D\\ \vdots & \vdots & \vdots & \vdots & \vdots & \vdots & \vdots & \vdots \\ S_{1,M} & C_{1,M} & S_{2,M} & C_{2,M} & \cdots & S_{P,M} & C_{P,M} & D \end{array}\right] \end{array}$$
where
(5)$$\begin{array}{cc} \begin{array}{c} \omega =2\pi f \end{array} & t_{m}=mT_{c}\\ S_{p,m}=\sin\left(p\omega t_{m}\right) & C_{p,m}=\cos\left(p\omega t_{m}\right) \end{array}$$


The over-determined system (Equation 3) has the following solution:(6)$$\hat{x}={\left[G^{T}G\right]}^{-1}G^{T}s$$

By taking into consideration the operation performed by the algorithm to solve Equation (6), it is possible to demonstrate that the computational complexity of this algorithm is **O**(n).

### 4.2. PQ Event Detection

According to the considered models, the PQ event generates harmonics, particularly in the transition phase. Several simulations are analyzed by considering signals affected by PQ events and signals not affected by PQ events to determine how the harmonics of the signal change in different cases. The results of the simulations highlight that, in the case of PQ events, the harmonics mainly affected are the first and the third ones. Due to the event typology, the fundamental component can increase or decrease its amplitude while the third harmonic component’s amplitude always increases.

As an example, Figure 5 shows the trend of a signal affected by Sag. The figure highlights the subset of the acquired samples in the SAW that are analyzed with the multi-sine fitting algorithm to detect the presence of PQ events. Figure 6 shows the amplitude of the fundamental component and of the third harmonic versus time, i.e., versus the sliding of the SAW. In the example, the SAW has a duration of 0.010 s, and it slides a number of samples Δs equal to 0.002 s. This figure highlights that it is possible to define threshold values to detect the presence of the event. Figure 7 shows an example of the trend of the first and third components for different typologies of PQ events with superimposed white Gaussian noise giving an SNR of 20 dB. From an exhaustive numerical investigation, it is highlighted that to detect the presence of any event, it is sufficient to verify that the fundamental-component normalized amplitude is out of the range of 0.9 and 1.1, and the estimated third-harmonic normalized amplitude is over 0.05.

### 4.3. Locally Distributed Node Management Algorithm

Unlike [19], for the continuous analysis of a time-varying signal that is affected by burst events, it is necessary to introduce appropriate acquisition and memory management algorithms. For this reason, in the LDN, two routines run in parallel modality. The first routine, described by Algorithm 1, manages the circular buffer (CB)’s recording procedure. In this algorithm, each newly acquired sample *q* is stored in the next memory location of the CB with respect to the previous sample. When the circular buffer is full, the oldest sample is overwritten by the newly acquired sample. This is possible by managing the specific memory index $I_{last}$ that, in its updating value, cannot overtake the dimension *N* of the circular buffer memory. The second routine detects if a PQ event occurs in the samples stored in the circular buffer. This routine is described in Algorithm 2. The index $I_{an}$ points to the location of the circular buffer containing the first samples to be analyzed and, therefore, where the SAW with length $N_{s}$ begins. If, in the circular buffer, there are at least $N_{s}$ new samples to be analyzed, then the samples from $I_{an}$ to $\mathrm{MOD}(I_{an}+N_{s},N)]$ are extracted, filtered, and analyzed to evaluate the amplitude of the fundamental and the third harmonic components. These two values are compared with threshold values to detect the possible presence of PQ events. Once this procedure is completed, the SAW slides by increasing the index $I_{an}$ of a number of samples equal to the sliding window increment Δs. A higher value of Δs decreases the computational burden but can lead to a miss-detection of the alteration. Therefore, the investigation to determine a suitable value of Δs is performed by numerical tests. Once an event is detected, the segmentation is performed by creating a set of samples suitable for the classification block. In particular, for the classifier proposed in [20], two periods of the signal before the event are included, and the total number of samples of the resulting segment is equal to 4000. This set of samples is transmitted to the CCU.
**Algorithm 1** Circular buffer management logicACQUIRE A NEW SAMPLE: *q*StORE *X* IN THE CIRCULAR BUFFER CB:     $I_{last}\leftarrow \mathrm{MOD}\left(I_{last}+1,N\right)$     $CB[I_{last}]\leftarrow X$

**Algorithm 2** Analysis algorithm

If $\mathrm{MOD}\left(I_{an}+N_{s},N\right)>I_{last}$:
   If $(I_{an}+N_{s})<N$:     $SW\leftarrow CB[I_{an}:I_{an}+N_{s}]$   Else     $SW\leftarrow CB\left[I_{an}:N\right],CB\left[0:\mathrm{MOD}\left(I_{an}+N_{s},N\right)\right]$   $H_{1},H_{2}\leftarrow MultisineAlg\left(SW\right)$   $I_{an}\leftarrow \mathrm{MOD}\left(I_{an}+\Delta s,N\right)$   RETURN $DetectAlteration(H_{1},H_{2})$


## 5. Results

The proposed method was preliminarily tested in different operating conditions by numerical tests. A code developed in Python was used to simulate different typologies of PQ events. For each typology, 100 signals were generated in different operating conditions as described in the following text. The sampling frequency used in all the tests was 10 kHz, and the duration of the signal acquired and stored in the circular buffer was equal to 0.4 s, which is the sampling frequency and signal length necessary for the selected classifier [20].

To establish the influence of the minimum number of samples $N_{s}$ of the SAW, allowing for the detection of the alteration in all cases, $N_{s}$ was varied in the range of [70, 180] samples. The signals were emulated with a superimposed white Gaussian noise with an SNR equal to 20 dB. In Figure 8, the percentages of correct detection for the different event typologies versus the number of samples $N_{s}$ are reported. The figure highlights that if the number of samples analyzed by the multi-sine fitting is greater or equal to the number of samples in a semi-period of the power signal, all the alterations are correctly detected. Therefore, increasing $N_{s}$ will not produce any advantages in the detection accuracy but will only increase the computation time. To optimize the execution of the proposed method on low-cost hardware with limited computational capability, a numerical test was designed to establish the Δs value. A higher value of Δs permits a reduction in the number of SAWs to be analyzed, but as said before, this can cause the missed detection of an alteration. In Figure 9 are shown the numerical results. The figure highlights that the events are correctly detected for a Δs equal or below to 40 samples, which is about a fifth of the input signal period. Subsequently, to minimize the number of samples processed by the three-parameter multi-sine fitting algorithm and, thus, to further reduce the computational time of the detection algorithm, other considerations were made related to the sampling frequency. The sampling frequency of the IoT measurement node was chosen according to the sampling frequency required by the classification algorithm. However, the number of samples used for the detection method can be reduced without decreasing its accuracy. To obtain this result, the acquired input signal can be down-sampled with a suitable decimation factor. In this case, the number of samples in the SAW $N_{s}$ can be reduced to $\tilde{N_{s}}$ without changing its time duration. The numerical tests show that for $N_{s}$ equal to or greater than 100 samples and Δs equal to or less than 40 samples, a decimation factor equal to 10 does not change the detection accuracy. A higher value of the decimation factor reduces the detection accuracy, in particular for the harmonic alteration.

Finally, to analyze the effect of white Gaussian noise on the detection accuracy, numerical tests were performed by varying the SNR in the range of [0, 40] dB. In Figure 10 are shown the numerical results. For an SNR value higher than 10 dB, all the PQ events were correctly detected. For lower SNR values, the sinusoidal signal was not correctly recognized. This was due to the increase in the noise floor, which also increased the harmonic components of the signal.

### Experimental Results

In order to evaluate the performance of the proposed detection method in a real scenario, a specific test bed was designed. Low-cost hardware with low computational capability was selected to implement the LDN. In particular, Algorithms 1 and 2 were implemented on the Arduino Due board. This board was based on the Atmel SAM3X8E ARM Cortex-M3 CPU. Its main characteristics were a clock frequency of 84 MHz and 12 analog inputs with an amplitude resolution of 12 bits. To emulate the power signal affected by PQ events, the National Instruments DAQ NI 6211 was used. The DAQ is characterized by an analog output with a 16 bits resolution and a maximum update rate of 250 kS/s. Figure 11 shows the block scheme of the measurement stand used in the experiment. The Arduino 2 analog input channel 1 is connected to the Analog Output of the DAQ. The digital pins of the Arduino 2 board are used to establish a serial connection with a communication module, which permits the transmission of the detected PQ events to the CCU. There are several wired and wireless solutions to enable communication between the LDN and the CCU. In this study, wireless communication systems were considered because they allow for a simple and flexible distribution of the LDNs, as they do not require cabling works. Among the different wireless communication protocols considered, some are reported in Table 1, which highlights their performance with respect to the data rate and transmission range. By considering a few events per hour, all of these standards can be used; otherwise, the standard with a higher data rate has to be considered. In the implemented measurement stand, a WiFi protocol was considered. In particular, the communication module was implemented with an ESP-01 board.

A PC with a suitable program developed in the LabView environment was used to manage the DAQ to generate the signals affected or unaffected by PQ events. The Arduino Due board was programmed to implement the proposed detection method and to send the detected PQ events to the PC that acts as a CCU. To evaluate the performance of the board, the computation time for each detection was sent to the CCU. Another PC was used as a CCU to collect the events detected by the Arduino Due board and transmit them with the WiFi Protocol.

Preliminary tests were conducted to determine the time required by the LDN to execute the proposed algorithms. In the tests, an $N_{s}$ equal to 100 was considered since, according to the numerical results (see Figure 8), this is the minimum value allowing the correct detection. The result shows that the LDN requires a time Δt of about 1000 μs to compute the parameters for a single SAW. With a sampling frequency equal to 10 kHz, the number of samples $N_{s}$ was acquired in 10,000 μs. Since the processing time is shorter than the acquisition time, it is possible to perform the detection and segmentation online with the acquisition. Implementing the parameters estimation method with the FFT instead of the multi-sine fitting algorithm, with the same hardware and operating conditions, the mean value of the execution time was 11,804 μs, which did not permit the online analysis because it was higher than the acquisition time.

To investigate the possibility of using hardware with lower computational performances for implementing the LDN, the execution time necessary to execute the proposed method versus $N_{s}$ was considered. In particular, the acquired samples were down-sampled before applying the fitting algorithm, obtaining $\tilde{N_{s}}$ samples. Table 2 shows the results for Δt and its standard deviation obtained with $\tilde{N_{s}}$ from 10 (down-sampling factor equal to 10) to 100 (down-sampling factor equal to 1). Δt also includes the time necessary to execute the down-sampling algorithm. The results highlight that by using a down-sampling factor of 2, the execution time for the evaluation of the harmonic amplitude in each SAW was reduced by 40%. However, by using a down-sampling factor of 10, the execution time was reduced to one-tenth. Subsequently, a down-sampling factor of 10 was used. To compare the experimental results with the simulation ones, 100 signals were generated for each alteration taken into consideration with a white Gaussian noise superimposed with an SNR equal to 20 dB. The detection algorithm was set to have an $N_{s}$ equal to 100 samples and a Δs equal to 25 samples. The experimental results confirmed the numerical ones with all the decimation factors.

## 6. Future Work

Different future works can be suggested for the proposed research and architecture. The main ones are as follows:The development of LDN prototypes with different communication standards, both wired and wireless;The characterization of the system even in the presence of more than one PQ events at the same time;The product engineering of the LDN;The development of the CCU, making it capable of classify the PQ events and evaluating the characteristic parameters of the recognized event [53,54,55];The development of a methodology and of a prototype suitable for three-phase systems. The literature analyzed shows few studies on unbalanced three-phase systems, in particular in the case of renewable energy sources [56].

## 7. Conclusions

The new power generation scenario increases power networks’ complexity and, consequently, the number of points to be monitored. This strongly increases the costs of the distributed monitoring system. Since high computational resources are required by the PQ event classification, in this paper, it is proposed to split the traditional monitoring node into two parts: the Locally Distributed Nodes (LDNs) and the Central Classification Unit (CCU). The LDN is installed on each monitoring point, and it is devoted to detecting PQ events and acquiring the segment of the power signal to be sent to a CCU that classifies the detected PQ events. As a consequence, one CCU can manage several LDNs. To strongly reduce the costs, the LDN must be implemented with low-cost hardware, and, thus, with reduced computation capabilities. For this purpose, a suitable PQ event detection and segmentation method is proposed. It is based on the multi-sine fitting algorithm and sliding window approach. The use of a multi-sine fitting algorithm achieved better performances with low-cost hardware compared with other decomposition methods. Numerical tests were performed to tune the acquisition parameters to increase the detection ratio, reduce the computational burden, and verify the suitability of the proposal. From the results of the numerical tests, it was highlighted that with a sliding window of at least 100 samples, PQ events were detected by the LDN in 100% of cases. The sliding window could slide up to 40 samples, keeping the detection percentage unchanged. Moreover, the method enabled us to operate up to 20 dB of SNRs, keeping the detection percentage unchanged. For SNR values lower than 20 dB, the proposed method detected the PQ events but had a false positive in the detection.

Experimental tests were executed to confirm the numerical ones in a real scenario. In these tests, the low-cost Arduino Due was used to implement the node. The down-sampling approach was used to further reduce the computational burden without losing detection effectiveness and, as a consequence, enabled the use hardware with lower computational capability as a node. In particular, the execution time with a sliding window of 100 samples required a mean execution time of 1000 μs instead of the parameter estimation obtained in the same conditions by the FFT that required 11,804 μs. By considering this, it is possible to assert that in order to not lose detection accuracy even in high-noise conditions, the proposed method can be used with a sliding window of 100 samples and a slide of less than 40 samples without requiring the use of high-performance hardware for the processing of the acquired signal.

## Figures and Tables

**Figure 1 sensors-24-02474-f001:**
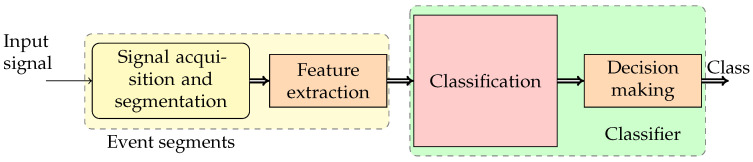
Block diagram of a standard automatic PQ event classifier [9,12].

**Figure 2 sensors-24-02474-f002:**
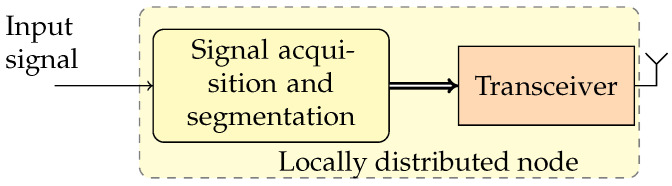
Block diagram of the proposed locally distributed node for the detection and segmentation of PQ events.

**Figure 3 sensors-24-02474-f003:**
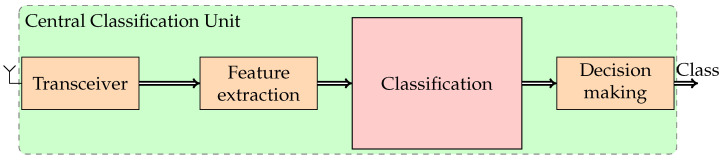
Block diagram of central classification unit of PQ events.

**Figure 4 sensors-24-02474-f004:**
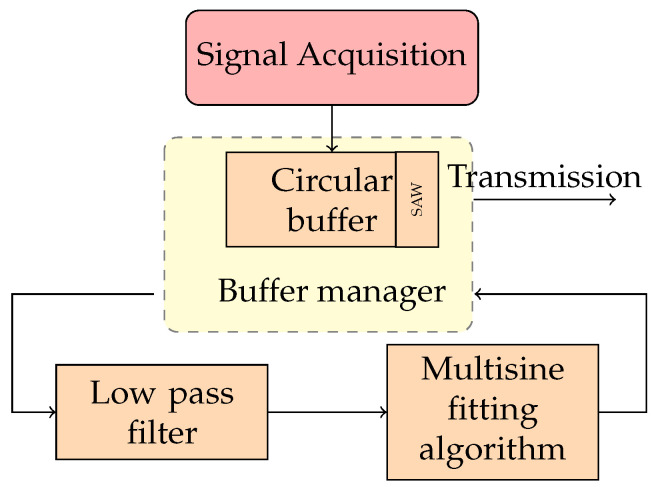
Block scheme of the proposed detection and segmentation algorithm.

**Figure 5 sensors-24-02474-f005:**
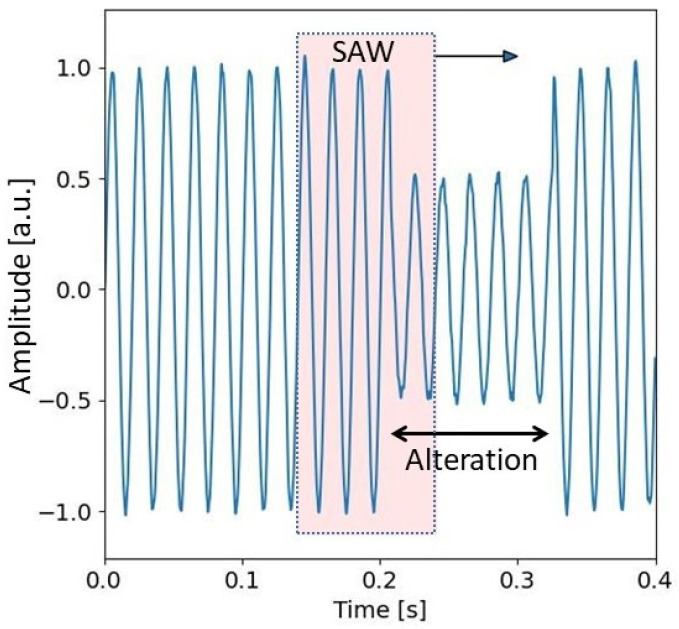
Example of signal affected by Sag event.

**Figure 6 sensors-24-02474-f006:**
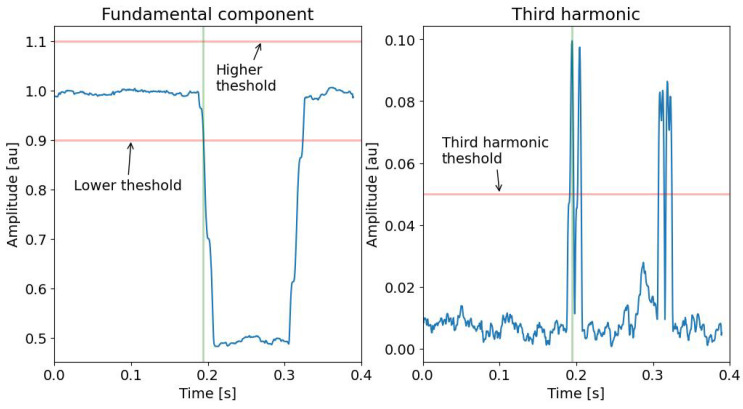
Fundamental and third harmonic amplitude trend of a Sag signal obtained by the proposed method.

**Figure 7 sensors-24-02474-f007:**
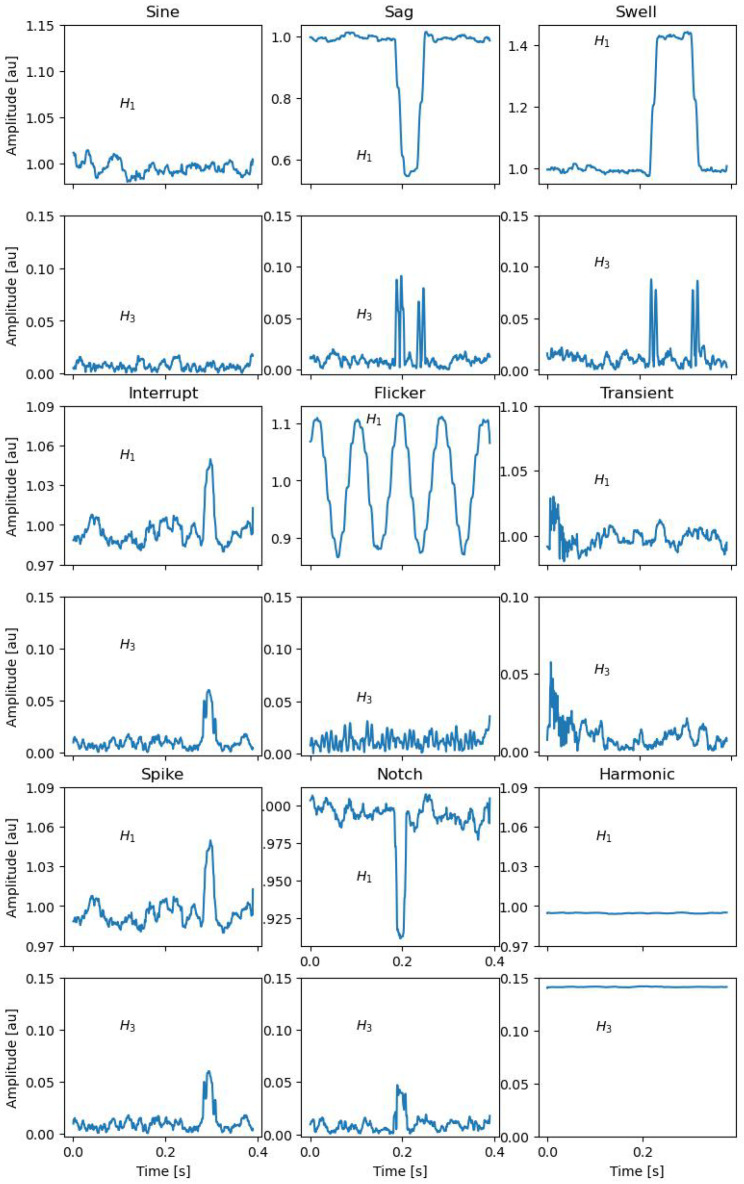
Example of $H_{1}$ and $H_{3}$ trend for different PQ events taken into consideration with an SNR equal to 20 dB.

**Figure 8 sensors-24-02474-f008:**
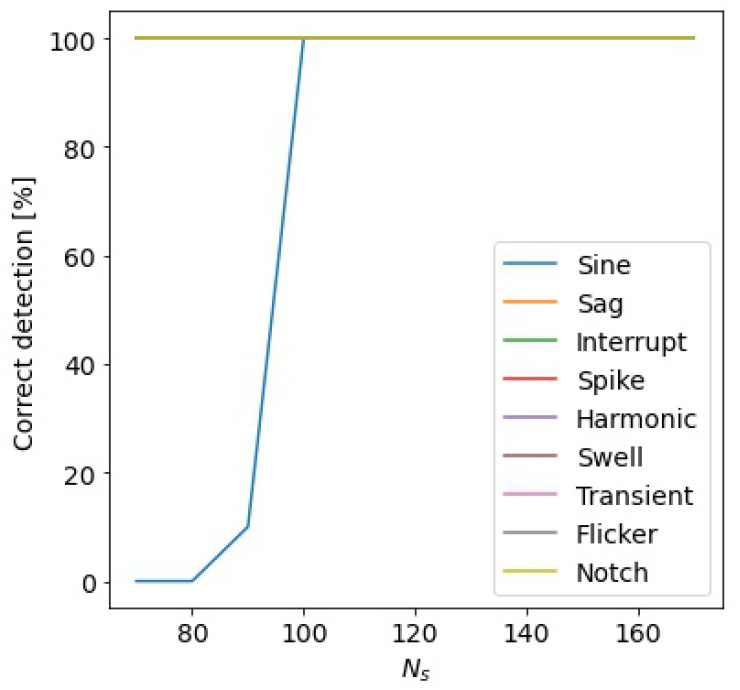
Percentage of correct detection of the alterations versus the number of samples of the sliding window $N_{s}$.

**Figure 9 sensors-24-02474-f009:**
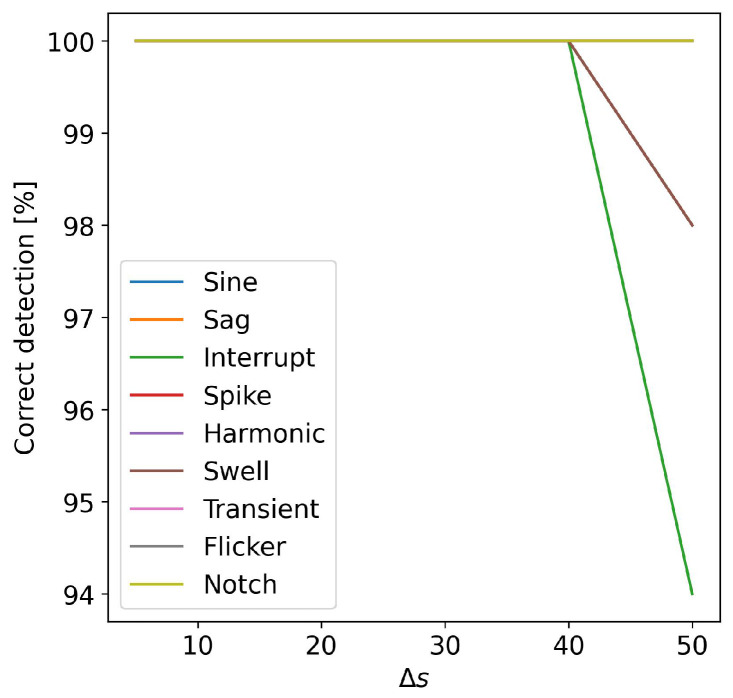
Percentage of correct detection of the alterations versus Δs.

**Figure 10 sensors-24-02474-f010:**
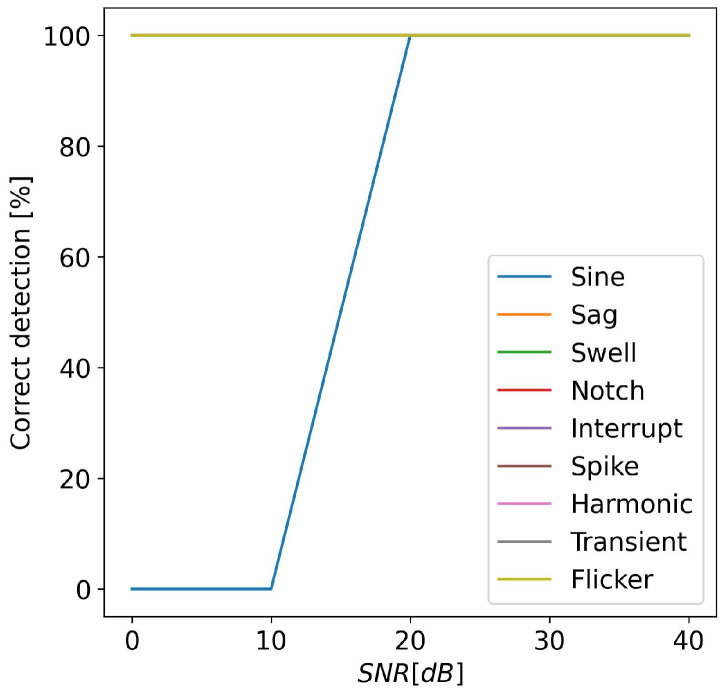
Percentage of correct detection of alterations versus SNR.

**Figure 11 sensors-24-02474-f011:**
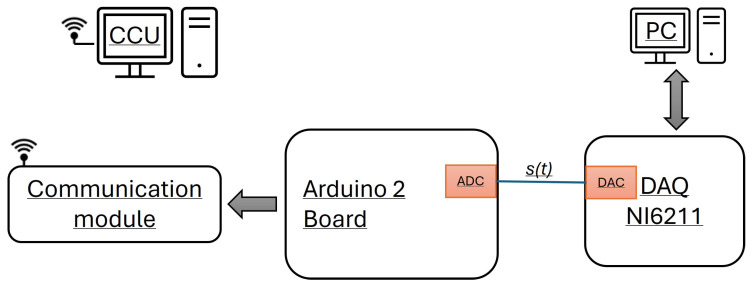
Block diagram of the measurement stand used for the experimental tests.

**Table 1 sensors-24-02474-t001:** Wireless communication alternatives.

Protocols	Data Rate	Range
Zigbee	250 kbps	up to 200 m
Bluetooth Low energy	1 Mbps	15–30 m
SigFox	100 bps UL	10 km urban
	600 bps DL	50 km rural
6lowpan	250 kbps	15–30 m
LoRaWan	29 bps–50 kbps	2–15 km
Z-wave	40–100 kbps	30 m (indoors)
		100 m (outdoors)
Bluetooth	1, 3, 24 Mbps	10–100 m
IEEE 802.11b	1, 2, 5.5, 11 Mbps	35–140 m
GPRS	56, 171.2 kbps	>1 km

**Table 2 sensors-24-02474-t002:** Experimental execution time of the sliding window evaluation versus $\tilde{N_{s}}$.

$\tilde{N_{s}}$	Mean(Δt) [μs]	Std(Δt) [μs]
10	100	15
25	300	18
50	600	34
100	1000	81

## Data Availability

The data presented in this study are available on request from the corresponding author.

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
