# Peer review of "Local Distributed Node for Power Quality Event Detection Based on Multi-Sine Fitting Algorithm"

_sensors, 2024, doi:10.3390/s24082474_

Round 1

Reviewer 1 Report

Comments and Suggestions for Authors

Thank you for the good work.

I find it astonishing that it has taken us so long to get there in this domain, since neither the problem (cost of analysing "everything" in detail only to find out that there is nothing of interest; cost of having high computational power everywhere even though most of the time, there is nothing to be done) nor the solution (e.g. triggering mechanisms in high-energy physics applications such as the LHC's detectors; dailing in to powerful computers from simple terminals in the 1970s) is new. It is very good these approaches get applied to PQ, finally. 

Would it be worth mentioning the good experience with these approaches in other domains? Would it facilitate the uptake of the proposal "It works/worked well elsewhere for ages, it is even the basis of research that some people got Nobel prizes for, so why don't we apply it to our field?"?

Little editorial details:

According to the established rules of typesetting (BIPM's SI brochure, ISO 80000, ...), mathematical constants (such as pi and e = 2.718...), well-defined functions (such as sin, cos), operators (such as T for transposed), unit symbols (such as Hz for Hertz and s for second) and descriptive text (such as last) are to be set upright. Variables (such as the index p in "p-th harmonic", line 160) are to be set in italic font. It makes reading much easier when conventions are followed. Especially since you use s for second (upright) and as a variable (italic) (sliding window increment Delta s). When following the typesetting conventions, there is no ambiguity.

Line 163, reference [8?]. Does the ? indicate that something went wrong?

Author Response

Request: Would it be worth mentioning the good experience with these approaches in other domains? Would it facilitate the uptake of the proposal "It works/worked well elsewhere for ages, it is even the basis of research that some people got Nobel prizes for, so why don't we apply it to our field?"?

Reply: The Authors would thank Reviewer#1 for his encouraging comment. We believe that the proposed method can be applied wherever we have a signal with a known mathematical model, and it is actually affected by distortions that can also be modelled.

Request: According to the established rules of typesetting (BIPM's SI brochure, ISO 80000, ...), mathematical constants (such as pi and e = 2.718...), well-defined functions (such as sin, cos), operators (such as T for transposed), unit symbols (such as Hz for Hertz and s for second) and descriptive text (such as last) are to be set upright. Variables (such as the index p in "p-th harmonic", line 160) are to be set in italic font. It makes reading much easier when conventions are followed. Especially since you use s for second (upright) and as a variable (italic) (sliding window increment Delta s). When following the typesetting conventions, there is no ambiguity.

Reply: The authors thank the Reviewer for the suggestion which will allow us to improve the readability of this and future works. The highlighted typos are fixed in the text and in all formulas.

Request: Line 163, reference [8?]. Does the ? indicate that something went wrong?

Reply: The Authors thank the Reviewer for the observation. The problem was related to typos in the latex file. The problem is fixed now.

Reviewer 2 Report

Comments and Suggestions for Authors

In the paper under review, the authors proposed the idea of improving the architecture of the Automatic Power Quality Events Classifier (APQEC), which differs from the known ones by splitting the traditional monitoring nodes into two parts – Locally Distributed Nodes (LDNs) and a Central Classification Unit (CCU). LLBs are low-cost hardware with reduced computation capabilities. CCU is a more computationally powerful device that implements complex classification algorithms. In general, the research presented by the authors represents the development of approaches to monitoring the quality of electricity in the power grid and therefore it is relevant and has important practical significance.

In the Introduction section, the authors presented a brief literature review that substantiated the relevance of the research topic and formulated the purpose of the paper. The main parts of the paper provide a description of the proposed distributed APQEC architecture, including possible algorithms for the operation of the CCU. The results of numerical tests under various operating conditions are presented, using a specially designed test bed implemented on the Arduino Due board and the LabView software.

In general, the presented paper will be of interest to scientists and researchers, specialists in the field of monitoring of power grid. However, I still have questions and the following comments:

1.     It seems to me that the Abstract does not fully reflect the content of the paper. I suggest the authors revise and expand it.

2.     I consider it necessary to note that the authors were unable to clearly formulate the scientific novelty of their research. The paper looks like a solution to a specific engineering problem. And this is not enough for publication in a highly-rated scientific journal. I ask the authors to make corrections to the paper and clearly formulate the scientific novelty.

3.     Throughout the text of the paper, references are provided in the form of hyperlinks. This does not correspond to the generally accepted format of scientific articles. Authors must remake references in the form “[1-3]”.

4.     A more detailed description of the algorithms proposed by the authors is required, with a clear description of the differences from the known developments referred to by the authors.

5.     Perhaps the test bed designed by the authors and used during experimental studies should be described in more detail.

6.     The Conclusion section should provide specific recommendations for applying the results obtained. In its present form, only a brief description of what is given in the paper is presented.

Therefore, I cannot recommend this paper for publication in the Sensors journal in its present form. The paper needs to be revised in light of the comments presented above.

Author Response

The Authors would thank the Reviewers for their precious advice to ameliorate the paper titled: “Local Distributed Node for Power Quality Event Detection Based on multi-sine fitting algorithm”. The paper is revised according to the Reviewers' comments and suggestions. In the revised version the modified text is highlighted in yellow from the introduction to the end, and the modified text of the abstract is highlighted by using red colour characters (we are sorry but there are some limitations in the template that do not allow to highlight the abstract).

In the following, there are the replies to the Reviewer's requests, and in the attached pdf file there is also the new image included. 

Reviewer 2:

In the paper under review, the authors proposed the idea of improving the architecture of the Automatic Power Quality Events Classifier (APQEC), which differs from the known ones by splitting the traditional monitoring nodes into two parts – Locally Distributed Nodes (LDNs) and a Central Classification Unit (CCU). LLBs are low-cost hardware with reduced computation capabilities. CCU is a more computationally powerful device that implements complex classification algorithms. In general, the research presented by the authors represents the development of approaches to monitoring the quality of electricity in the power grid and therefore it is relevant and has important practical significance.

In the Introduction section, the authors presented a brief literature review that substantiated the relevance of the research topic and formulated the purpose of the paper. The main parts of the paper provide a description of the proposed distributed APQEC architecture, including possible algorithms for the operation of the CCU. The results of numerical tests under various operating conditions are presented, using a specially designed test bed implemented on the Arduino Due board and the LabView software.

In general, the presented paper will be of interest to scientists and researchers, specialists in the field of monitoring of power grid. However, I still have questions and the following comments:

Request 1:     It seems to me that the Abstract does not fully reflect the content of the paper. I suggest the authors revise and expand it.

Reply: as requested, the Abstract is revised according to the Reviewer request and the Sensor template recommendation. In particular, in the abstract, the following points are briefly introduced:

(1) Background: “The new power generation systems, the increasing number of equipment connected to the power grid, and the introduction of technologies such as the smart grid, underline the importance and complexity of the Power Quality (PQ) evaluation.

In this scenario, an Automatic PQ Events Classifier (APQEC) that detects, segments, and classifies the anomaly in the power signal is needed for the timely intervention and maintenance of the grid. Due to the extension and complexity of the network, the number of points to be monitored is huge making the cost of the infrastructure unreasonable.”

(2) Methods: “To reduce the cost, a new architecture for an APQEC is proposed. This architecture is composed of several Locally Distributed Nodes (LDNs) and a Central Classification Unit (CCU). The LDNs are in charge of the acquisition, the detection of PQ events, and the segmentation of the power signal. Instead, the CCU receives the information from the LDNs to classify the PQ events.

A low-computational capability characterizes low-cost LDNs. For this reason, a suitable PQ event detection and segmentation method with low computational resource requirements is proposed. It is based on the use of a sliding observation window that establishes a reasonable time interval, also useful for signal classification and the multi-sine fitting algorithm to decompose the input signal in harmonic components. These components can be compared with established threshold values to detect if a PQ event occurs. Only in this case, the signal is sent to the CCU for the classification, otherwise, it is discarded.”

(3) Results: “Tests are performed to set the sliding window size and observe the behaviour of the proposed method in the case of the main PQ events reported in the literature, even when the SNR varies.”

(4) Conclusions: “Experimental results confirm the effectiveness of the proposal highlighting the correspondence with numerical results and the reduced execution time when compared to FFT-based methods”.

As a consequence, the Abstract is rewritten as follows:

“The new power generation systems, the increasing number of equipment connected to the power grid, and the introduction of technologies such as the smart grid, underline the importance and complexity of the Power Quality (PQ) evaluation. In this scenario, an Automatic PQ Events Classifier (APQEC) that detects, segments, and classifies the anomaly in the power signal is needed for the timely intervention and maintenance of the grid. Due to the extension and complexity of the network, the number of points to be monitored is huge making the cost of the infrastructure unreasonable. To reduce the cost, a new architecture for an APQEC is proposed. This architecture is composed of several Locally Distributed Nodes (LDNs) and a Central Classification Unit (CCU). The LDNs are in charge of the acquisition, the detection of PQ events, and the segmentation of the power signal. Instead, the CCU receives the information from the nodes to classify the PQ events. A low-computational capability characterizes low-cost LDNs. For this reason, a suitable PQ event detection and segmentation method with low resource requirements is proposed. It is based on the use of a sliding observation window that establishes a reasonable time interval, also useful for signal classification and the multi-sine fitting algorithm to decompose the input signal in harmonic components. These components can be compared with established threshold values to detect if a PQ event occurs. Only in this case, the signal is sent to the CCU for the classification, otherwise, it is discarded. Numerical tests are performed to set the sliding window size and observe the behaviour of the proposed method with the main PQ events presented in the literature, even when the SNR varies. Experimental results confirm the effectiveness of the proposal highlighting the correspondence with numerical results and the reduced execution time when compared to FFT-based methods .”

Request 2:     I consider it necessary to note that the authors were unable to clearly formulate the scientific novelty of their research. The paper looks like a solution to a specific engineering problem. And this is not enough for publication in a highly-rated scientific journal. I ask the authors to make corrections to the paper and clearly formulate the scientific novelty.

Reply: The Authors thank the reviewer for highlighting this point. The scientific relevance of the paper consists in the problem itself that for many years is taken by the scientific communities, especially in the fields of metrology, computer science, telecommunications, and electronics, i.e. the optimization of the computational resources and the reduction of the costs. The scientific novelties of the paper are two. the approach, for the first time it is proposed to differentiate the measurement nodes in two subsets with different tasks and computational needs. The algorithm for the detection of the event. It is based on previous research of the authors but modified for the specific case, as described in the newly added text.

To highlight these points in the revised version of the paper the Introduction is modified as follows:

“……
The PQ evaluation is typically performed after a problem occurs by a trained operator. The reliability of the evaluation strongly depends on the operator’s ability and experience. Moreover, this approach does not permit preventive actions. Furthermore, the probability that a PQ event occurs during the operator’s observation time can be low, making the monitoring ineffective. For these reasons, recently, several devices for the automatic detection and classification of PQ events were presented in literature [ 6 ]. The general architecture [ 7-9] is shown in Fig.1 and it is composed of the blocks: signal acquisition and segmentation, Feature extraction, Classification, and Decision making. The relevant cost of the devices based on this architecture mainly depends on the computational requirements of the classifications block. High costs limit the scalability of the monitoring system and its capillary distribution over the whole power network. Moreover, due to the source of the PQ events, they can occur randomly, and it is not always possible to define an exact time in which an event will occur [ 11]. For this reason, the monitoring and classification system will, most of the time, do nothing except wait for an event, as long as an event happens, and its significant computing power will remain practically unused for almost all of the time. It is worth noting that not all the operations need to be executed at the measurement point. In particular, only the detection and acquisition of the PQ event must be performed at the measurement point, the classification can be performed by a remote device or in the Cloud.

For this reason, in this paper, a new Automatic PQ Events Classifier (APQEC) architecture is proposed. It is made up of two main component sets, the first is a set of Locally Distributed Nodes (LDN) that detect, segment, and acquire the portion of the Power Signal (PS) affected by the PQ event. These data are then transmitted to the second set of Central Classification Units (CCUs) which perform the PQ event classification. The advantage of this architecture is that the LDN can be implemented with low-cost devices, and are the only ones that need to be distributed, while each CCU can manage multiple LDNs, thus reducing the overall cost of the proposed monitoring system.

The improvement of the proposed solution is highlighted when the number of measurement points increases. Considering that tools such as Power Quality Analyzers or PCs and hardware for the implementation of the methods proposed in the literature cost more than a thousand Euros, this cost must be multiplied by the measurement points. Instead, in the proposed solution a single PC allows you to monitor multiple measurement points with LDNs which cost a few tens of Euros.

The scientific novelty of this approach in which a remote low-cost and low-capability subsystem triggers the local high-capability subsystem, it can be used in completely different areas than PQ, such as intrusion detection and classification, telecommunication signal detection and classification, Structural Health Monitoring system, and so on. From the analysis of the recent literature, the proposed methods perform the detection as the result of the classification [9] or use mathematical tools that are computationally expensive [10] and therefore all of them are not suitable for implementation on HW with reduced computational capabilities. As a consequence, since the aim of the paper is to reduce the monitoring costs by proposing a LDN with limited computational resources, a new detection and segmentation method characterized by a lower computational burden is proposed. The proposed detection method uses the multi-sine fitting algorithm [17] to recognize the presence of PQ events. Differently from [17 ] a Sliding Analysis Window (SAW) is used to permit the analysis of a time-varying signal and determine the input of the multi-sine fitting algorithm…….”

Request 3:     Throughout the text of the paper, references are provided in the form of hyperlinks. This does not correspond to the generally accepted format of scientific articles. Authors must remake references in the form “[1-3]”.

Reply: In the revised version of the paper the references follow the Reviewer's recommendations.

Request 4:     A more detailed description of the algorithms proposed by the authors is required, with a clear description of the differences from the known developments referred to by the authors.

Reply: The multi-sine fitting algorithm, as reported in the text of the work, it is the one included in the references of the article. The main difference with this is the management of the acquisition and analysis of the results which is specific to the problem we want to solve. To clarify this aspect in the article subsections 4.2 and 4.3 have been modified in the revised article. For example, the following sentence is included in the subsection 4.2:

“Differently from [17] for the continuous analysis of a time-varying signal that is affected by burst events, it is necessary to introduce appropriate acquisition and memory management algorithms. For this reason, in the LDN, two routines run in parallel modality. The first routine, described by algorithm 1, manages the circular buffer (CB) ……….”

Request 5:     Perhaps the test bed designed by the authors and used during experimental studies should be described in more detail.

Reply: As requested by the Reviewer more details about the measurement stand are included in the revised paper in subsection 5.1. In particular, a new figure and the following sentences are included in the paper:

“In Fig.11 it is shown the block scheme of the measurement stand used in the experiment. The Arduino 2 analog input channel 1 is connected to the Analog Output of the DAQ. The digital pins of the Arduino 2 board are used to establish a serial connection with a communication module, that permits the transmission of the detected PQ events to the CCU. There are several wired and wireless solutions to permit communication between the LDN and the CCU. In this paper wireless communication systems are considered because they allow simple and flexible distribution of the LDNs as they do not require cabling works. Among the different wireless communication protocols considered there are that reported in Tab.1 that highlights their performance with respect to data rate and transmission range. By considering a few events per hour all these standards can be used, otherwise, the standard with higher data-rate has to be considered. In the implemented measurement stand a WiFi protocol is considered. In particular, the communication module is implemented with an ESP-01 board.

A PC with a suitable program developed in LabView environment is used to manage the DAQ to generate the signals affected or not by PQ events. The Arduino Due board is programmed to implement the proposed detection method and to send the detected PQ events to the PC acts as CCU. To evaluate the performance of the board, the computation time for each detection is sent to the CCU. A further PC is used as CCU to collect the events detected by the Arduino Due board and transmitted with the WiFi Protocol.

Figure 11. Block diagram of the measurement stand used for the experimental tests

Request 6:     The Conclusion section should provide specific recommendations for applying the results obtained. In its present form, only a brief description of what is given in the paper is presented.

Reply: In the revised paper the Conclusion section includes the Reviewer recommendation. In particular, it is enriched as follows:

From the results of the numerical tests it is highlighted that with a sliding window of at least 100 samples, PQ events are detected by the LDN in 100% of cases. The sliding window can slide up to 40 samples keeping the detection percentage unchanged. Moreover, the method permits to operate up to 20 dB of SNR keeping the detection percentage unchanged, for SNR values lower than 20 dB the proposed method detects the PQ events but has false positive in the detection.

Experimental tests are executed to confirm the numerical ones in a real scenario. In these tests, the low-cost Arduino Due is used to implement the node. The downsampling approach can be used to further reduce the computational burden without losing detection effectiveness and, as a consequence, use hardware with lower computational capability as a node. In particular, the execution time with a sliding window of 100 samples requires a mean execution time of 1000 ms, instead of the parameter estimation obtained in the same conditions by the FFT that requires 11804 ms. By considering this, it is possible to assert that in order to not lose detection accuracy even in high noise conditions, the proposed method can be used with a sliding window of 100 samples and a slide of less than 40 samples without requiring the use of high-performance hardware for the processing of the acquired signal.”

.

Reviewer 3 Report

Comments and Suggestions for Authors

1. There are many literatures focus on power quality event detection. The reviewer recommends that this manuscript conduct in-depth discussion and analysis with more related literatures to enhance contribution and novelty.

2. This manuscript proposes several locally distributed nodes and a central classification unit architecture. The reviewer recommends that this architecture should be explained in detail to strengthen the differences in necessity and practicality. Otherwise, there are no different with commonly used machine edge computers and cloud platform architectures.

3. What is the hardware and communication technology of transceiver used in this manuscript? How does this manuscript deal with issues such as communication interference and transmission errors? Do the above issues affect the prediction classification mechanism?

4. Deep learning recognition model training and identification have been carried out for power quality event detection. The reviewer recommends that this paper conduct experimental performance analysis and discussion with relevant literature, such as [A] and [B].

5. There are some hyperparameter values within the system (such as the s). What was the strategy of fine-tuning them (and what is the impact of their values on the overall capabilities of the system?).

6 How about the computation complexity of the proposed method compared with related work? The performance comparison to other improved schemes is required.

7. In the conclusions section, the low-cost Arduino Due is used to implement the locally distributed node. The system architecture, system design and system implementation should be described more detial in the section 2.

8. The conclusion and future work part can be extended to have a better understanding of the approach and issues related to that which can be taken into consideration for future work.

[A] N. Rodrigues, F. Janeiro and P. Ramos, Deep Learning for Power Quality Event Detection and Classification Based on Measured Grid Data, IEEE Transactions on Instrumentation and Measurement, vol. 72, pp. 1-11, 2023.

[B] I. Topaloglu, Deep Learning Based a New Approach for Power Quality Disturbances Classification in Power Transmission System, Journal of Electrical Engineering and Technology, vol. 18, pp. 77-88, 2023.

Author Response

The Authors would thank the Reviewers for their precious advice to ameliorate the paper titled: “Local Distributed Node for Power Quality Event Detection Based on multi-sine fitting algorithm”. The paper is revised according to the Reviewers' comments and suggestions. In the revised version the modified text is highlighted in yellow from the introduction to the end, and the modified text of the abstract is highlighted by using red colour characters (we are sorry but there are some limitations in the template that does not allow to highlight the abstract).

In the following, there are the replies to the Reviewer's requests. In the file, there are also the new figure and the highlighted new text included in the revised paper.  

Reviewer 3

Request 1: There are many literatures focus on power quality event detection. The reviewer recommends that this manuscript conduct in-depth discussion and analysis with more related literatures to enhance contribution and novelty.

Reply: The authors agree that the number of articles in this field is huge demonstrating the interest and the importance of this topic in the scientific community. However, from the analysis of the literature focused on the detection of PQ events, it arises that many proposed detection methods are both for detection and classification, i.e. requires high computational capabilities. Indeed, the detection is performed by classifying the signal, this requires a huge computational and memory capabilities, i.e. expensive hardware. The few methods devoted only to detect the PQ events are based on complex mathematical approached that need high computation capabilities and, as a consequence, cannot be implemented on low-cost hardware. To highlight these aspects the introduction and the reference of the revised paper are modified as follow:

From the analysis of the recent literature, the proposed methods  perform the detection as the result of the classification [9] or use mathematical tools that are computationally expensive [10] and therefore all of them are not suitable for implementation on HW with reduced computational capabilities. As a consequence, since the aim of the paper is to reduce the monitoring costs by proposing a LDN with limited computational resources, a new detection and segmentation method characterized by a lower computational burden is proposed.

Among the reference there are:

“9. Mahela, O.P.; Shaik, A.G.; Gupta, N. A critical review of detection and classification of power quality events. Renewable and Sustainable Energy Reviews 2015, 41, 495–505. https://doi.org/https://doi.org/10.1016/j.rser.2014.08.070.
10. Ucar, F.; Alcin, O.F.; Dandil, B.; Ata, F. Power Quality Event Detection Using a Fast Extreme Learning Machine. Energies 2018, 11.”

Request 2: This manuscript proposes several locally distributed nodes and a central classification unit architecture. The reviewer recommends that this architecture should be explained in detail to strengthen the differences in necessity and practicality. Otherwise, there are no different with commonly used machine edge computers and cloud platform architectures.

Reply: The Authors thank the Reviewer to arise this question. The proposed solution cannot be considered a form of edge computing as it does not have the purpose of carrying out signal processing on the node to minimize the data transferred to the processing centre. The innovative idea described in the paper is to divide a measurement instrument into two different systems, the first one (that can be replicated in several places) is low cost and in charge only for the acquisition, detection and transmission of the event and positioned at the measurement point. The second one is remote and analyses the transmitted signal. Cloud architecture can be used to implement a future version of the proposed distributed measurement system, but it will be the objective of future research. In order to highlight this point, the introduction and section 5 are modified to clarify these aspects.

Request 3: What is the hardware and communication technology of transceiver used in this manuscript? How does this manuscript deal with issues such as communication interference and transmission errors? Do the above issues affect the prediction classification mechanism?

Reply: In this paper, an ESP01 is used for the WiFi communication of the LDN with the CCU, and the problems related to the communication interference are those of the standard IEEE 802.11. The authors want to highlight that the structure for developing the prototype is modular and therefore the communication module can be easily replaced without modifying the analysis methodology or the node itself. For the prototype, WiFi communication was chosen to make easier the implementation of the laboratory stand. However, other wireless protocols can be easily used, and some suggestions are included in section 5 of the revised paper. Even wired protocols can be used with the disadvantage of requiring much more expensive installation work than wireless ones. All these aspects are now described in the revised section 5. In particular, a new figure and the following sentences are include in the paper:

In Fig.11 it is shown the block scheme of the measurement stand used in the experiment. The Arduino 2 analog input channel 1 is connected to the Analog Output of the DAQ. The digital pins of the Arduino 2 board are used to establish a serial connection with a communication module, that permits the transmission of the detected PQ events to the CCU. There are several wired and wireless solutions to permit communication between the LDN and the CCU. In this paper wireless communication is considered because it allows the simple and flexible distribution of the LDNs as it does not require cabling works. Among the different wireless communication protocols taken into account there are that reported in Tab.1 that highlights their performance with respect to data rate and transmission range. By considering a few events per hour all these standards can be used, otherwise, the standard with higher data-rate has to be considered. In the implemented measurement stand a WiFi protocol is considered. In particular, the communication module is implemented with an ESP-01 board.

A PC with a suitable program developed in LabView environment is used to manage the DAQ. It is used to generate signals affected or not by PQ events. The Arduino Due board is programmed to implement the proposed detection method and to send the detected PQ events to the PC actin as CCU. In order to evaluate the performance of the board, the computation time for each detection is sent to the CCU. A further PC is used as CCU to collect the events detected by the Arduino Due board and transmitted with the WiFi Protocol.

Figure 11. Block diagram of the measurement stand used for the experimental tests

Request 4: Deep learning recognition model training and identification have been carried out for power quality event detection. The reviewer recommends that this paper conduct experimental performance analysis and discussion with relevant literature, such as [A] and [B].

Reply: The Authors thank the Reviewer for the suggestions. Both papers are analyzed and are now included in the bibliography. A comparison with the proposed method is not possible, because the detection method implemented by these actually performs classification of all the incoming signals. So they can not be implemented on LDN.

Request 5: There are some hyperparameter values within the system (such as the s). What was the strategy of fine-tuning them (and what is the impact of their values on the overall capabilities of the system?).

Reply: The effect of the parameters such as the slide of the observation window (Ds) and the number of samples of the sliding observation window (Ns) are analyzed in Section 5. In particular, this section highlights that for a Ds up to 40 samples the percentage of correct detection is equal to 100%. Moreover, 100 samples of sliding window are necessary to correctly detect the sinusoidal signal, otherwise it is detected as a PQ event. Some recommendations are included in the conclusion of the revised paper.

Request 6: How about the computation complexity of the proposed method compared with related work? The performance comparison to other improved schemes is required.

Reply: From the analysis of the literature conducted by the Authors there are no methods that perform the detection without the classification or complex mathematical tool. Instead, the detection is obtained by classifying the signal, and is not possible to implement these methods on the LDN. However, as regards the extraction of the information necessary for the detection with the proposed method, the comparison with the classic FFT is in the work.

Request 7: In the conclusions section, the low-cost Arduino Due is used to implement the locally distributed node. The system architecture, system design and system implementation should be described more detial in the section 2.

Reply: More details about the LDN and the measurement test bed used to test the LDN and the proposed methodology are included in subsection 5.1 as reported in the reply of the request 3.

Request 8: The conclusion and future work part can be extended to have a better understanding of the approach and issues related to that which can be taken into consideration for future work.

Reply: As requested by the Reviewer the conclusion is improved and the future work section is included. In particular, the new section Future work is as follows:

“6. Future work 

Different future works can be suggested for the proposed research and architecture.

The main regard:

  • the development of LDN prototypes with different communication standards, both wired and wireless,
  • the characterization of the system even in the presence of more than one PQ event at the same time,
  • the product engineering of the LDN,
  • the development in the CCU, capable to classify the PQ events and evaluate the characteristic parameters of the recognized event [51–53],
  • the development of a methodology and of a prototype suitable for three-phase system. The literature analysed gives few studies about unbalanced three-phase systems, in particular in the case of renewable energy sources [54].”

Reviewer 4 Report

Comments and Suggestions for Authors

The topic addressed by the authors, in this paper, is a very current one, which requires an ever-increasing advance.

Distributed generation makes our systems and networks even more complex, both hardware and software. This complexity comes with new challenges, therefore also with great opportunities.

With the emergence of the new paradigm, distributed generation, the quality of electrical energy was strongly affected. There were problems before, but with this paradigm shift, these problems grew exponentially. The complexity is increasing, obviously we need more and more monitoring points, more and more data and many pertinent analyses. This is so that we can respond in a timely manner to the problems that arise. Through their proposal, the authors fit into the current concerns in this direction.

The Automatic PQ Events Classifier architecture proposed by the authors in this paper is an interesting one, presented in a well-structured way. The approach of the authors for an architecture in which the separation of the traditional monitoring node into two parts seems, from partial results, to bring advantages to the proposed methodology.

PQ event detection and segmentation method based on the multi-sine fitting algorithm and Sliding Window approach brings a series of other benefits to the methodology proposed by the authors. Increased performance with a lower price hardware. The results obtained by the numerical methods show the robustness of the proposal and encourages for other tests. Tests in real conditions also bring satisfactory results. The conclusions presented by the authors are supported by the presented results.

In order to increase the scientific value of the work, I would have the following observations:

1. when you say "low-cost", I think it would be good to present what this means financially and roughly where it stands on the market with that price compared to other solutions. A comparison is welcome, to support other statements.

2. the bibliographic list contains very few works from the last 3-4 years. Somewhere 22% of the references are current, which does not give sufficient credibility to the Introductory part (where we should compare our ideas with the ideas of the last periods, 2, maximum 3 years), or to the list of references. And this lowers the quality of the work, even if the idea is good and presented well.

I think that with these minor changes, the paper can acquire the desired scientific value.

Author Response

Cover letter

The Authors would thank the Reviewers for their precious advice to ameliorate the paper titled: “Local Distributed Node for Power Quality Event Detection Based on multi-sine fitting algorithm”. The paper is revised according to the Reviewers' comments and suggestions. In the revised version the modified text is highlighted in yellow from the introduction to the end, and the modified text of the abstract is highlighted by using red colour characters (we are sorry but there are some limitations in the template that does not allow to highlight the abstract).

In the following, there are the replies to the Reviewer's requests.

sincerely,

Francesco Lamonaca

Reviewer 4

The topic addressed by the authors, in this paper, is a very current one, which requires an ever-increasing advance.

Distributed generation makes our systems and networks even more complex, both hardware and software. This complexity comes with new challenges, therefore also with great opportunities.

With the emergence of the new paradigm, distributed generation, the quality of electrical energy was strongly affected. There were problems before, but with this paradigm shift, these problems grew exponentially. The complexity is increasing, obviously we need more and more monitoring points, more and more data and many pertinent analyses. This is so that we can respond in a timely manner to the problems that arise. Through their proposal, the authors fit into the current concerns in this direction.

The Automatic PQ Events Classifier architecture proposed by the authors in this paper is an interesting one, presented in a well-structured way. The approach of the authors for an architecture in which the separation of the traditional monitoring node into two parts seems, from partial results, to bring advantages to the proposed methodology.

PQ event detection and segmentation method based on the multi-sine fitting algorithm and Sliding Window approach brings a series of other benefits to the methodology proposed by the authors. Increased performance with a lower price hardware. The results obtained by the numerical methods show the robustness of the proposal and encourages for other tests. Tests in real conditions also bring satisfactory results. The conclusions presented by the authors are supported by the presented results.

In order to increase the scientific value of the work, I would have the following observations:

Request 1: when you say "low-cost", I think it would be good to present what this means financially and roughly where it stands on the market with that price compared to other solutions. A comparison is welcome, to support other statements.

Reply: The Authors have analysed the market and a power quality analizer (such as the Fluke 1770) costs over € 8000. The solutions proposed in the literature typically have the cost of a medium-performance PC plus the one of a data acquisition system for each measurement point (about €1500). Our solution requires only one medium-performance PC (about € 800) and an LDN for each measurement point (the cost of our LDN prototype is about € 25).  As a consequence, the advantages of our proposal are still valid in the case of one measurement point but strongly increase with the increasing number of measurement points (that is the nowadays scenario). The authors would thank the Reviewer for allowing us to better highlight this aspect in the introduction of the revised paper.

Request 2: the bibliographic list contains very few works from the last 3-4 years. Somewhere 22% of the references are current, which does not give sufficient credibility to the Introductory part (where we should compare our ideas with the ideas of the last periods, 2, maximum 3 years), or to the list of references. And this lowers the quality of the work, even if the idea is good and presented well.

Reply: The Authors would thank the reviewer for the opportunity to improve their work. Based on the Reviewer's request and to reply to the other Reviewers' requests in the revised paper almost 33 % of the references have been published since 2022, and almost 26 % have been published since 2023.

Round 2

Reviewer 2 Report

Comments and Suggestions for Authors

The authors provided detailed responses to my comments and expanded the paper. In its new form, I can recommend it for publication in the Sensors journal.

Author Response

The Authors would like to thank the Reviewer for his valuable contribution in improving the paper. 

Reviewer 3 Report

Comments and Suggestions for Authors

While I'm glad to see the author try to address my concerns, some of these issues remain unresolved.

1. The authors respond that "the few methods used to detect PQ events are based on complex mathematical methods that require high computational power and therefore cannot be implemented on low-cost hardware." . How computationally complex or low cost is the proposed method compared to related work?

2. The introduction and section 5 have not been revised to clarify my previous claim 2 (framework).

Author Response

The replies for the Reviewer are in the attached file.

Round 3

Reviewer 3 Report

Comments and Suggestions for Authors

This paper has edited and revised according to the reviewer's suggestions.